



# Dynamics of fortnightly water level variations along a tide-dominated estuary with negligible river discharge

Erwan Garel[1], Ping Zhang[2], Huayang Cai[2]

[1] Centre for Marine and Environmental Research (CIMA), University of Algarve, Faro, Portugal

[2] Institute of Estuarine and Coastal Research, School of Marine Engineering and Technology, Sun Yat-sen University, Guangzhou 510275, China

*Correspondence to:* Huayang Cai (caihy7@mail.sysu.edu.cn)

**Abstract.** Observations indicate that the fortnightly fluctuations in mean water level increase in amplitude along the lower

half of a tide-dominated estuary (The Guadiana estuary) with negligible river discharge but remain constant upstream. Analytical solutions reproducing the semi-diurnal wave propagation shows that this pattern results from reflection effects at the estuary head. The phase difference between velocity and elevation increases from the mouth to the head (where the wave has a standing nature) as the high and low water levels get progressively closer to slack water. Thus, the tidal (flood-ebb) asymmetry in discharge is reduced in the upstream direction. It becomes negligible along the upper estuary half, as the mean

sea level remains constant despite increased friction due to wave shoaling. Observations of a flat mean water level along a significant portion of an upper estuary, easier to obtain than the phase difference, can therefore indicate significant reflection of the propagating semi-diurnal wave at the head. Details of the analytical model shows that changes in the mean depth or length of semi-arid estuaries, in particular for macrotidal locations, affect the fortnightly tide amplitude, and thus the upstream mass transport and inundation regime. This has significant potential impacts on the estuarine environment.

## 1 Introduction

When averaged over a tidal cycle, the slope of the free surface elevation is generally not flat everywhere in an estuary. Several factors operating at distinct frequencies may be responsible for mean (or subtidal, tidally-averaged) water level variations along the channel, such as the tide, freshwater inputs, local or remote atmospheric conditions (e.g., wind, air pressure) and various coastal ocean processes acting at the mouth (e.g., Aubrey and Speer, 1985; Gallo and Vinzon, 2005; Henrie and Valle-

Levinson, 2014; Jay et al., 2015; Laurel-Castillo and Valle-Levinson, 2020; Matte et al., 2013; Ross and Sottolichio, 2016; Shetye and Vijith, 2013; Wong et al., 2009). At a fortnightly time scale, tide-dominated estuaries commonly feature relatively high and low mean water levels (MWL) on spring and neap tides, respectively, in relation to the tidal forcing variability produced by the interaction of the semidiurnal $M_2$ and $S_2$ tidal constituents (Aubrey and Speer, 1985). The resulting compound tide (MSf) has a 14.8-day period which corresponds to the beat period between $M_2$ and $S_2$ (i.e., $\omega S_2 - \omega M_2 = \omega MSf$, where $\omega$ is

the tidal frequency; Dworak and Gomez-Valdes, 2005; LeBlond and Mysak, 1978). The tidal potential also contains energy at





the MSf frequency, but this energy source is usually weak and becomes negligible in the upper estuary (LeBlond, 1979). Water level oscillations with MSf period along estuaries are typically referred to as fortnightly tides.

Fortnightly tides have been mainly described at long tidal rivers affected by substantial freshwater inputs. Subtidal changes in water elevation at these systems can be of metric order at the upper reaches and may have as such significant effects on

navigability and flood control management (e.g., Aubrey and Speer, 1985; Godin, 1999; Guo et al., 2015; Jay et al., 2015; Matte et al., 2014). MSf tides with large amplitude derive from the subtidal friction produced by the interaction between the river flow and the various tidal constituents (Buschman et al., 2009; Cai, 2016; Jay and Flinchem, 1997; Sassi and Hoitink, 2013). The river discharge nonlinearly enhances the subtidal friction experienced by the barotropic tidal wave propagating upstream (Godin, 1985; Godin, 1999; Guo et al., 2015; Jay, 1991; Matte et al., 2013; Pan et al., 2018; Sassi and Hoitink, 2013).

Due to its nonlinear dependence to water depth, the subtidal friction is greatest on spring tides and is thus balanced by the largest subtidal water level gradient landward; likewise, the gradient is smallest on neap tides when friction is comparatively weak. This is demonstrated by analytical derivations of the along-estuary momentum equation for tidal averaged conditions, which indicates that the water slope term is dominantly balanced by the friction term (e.g., Buschman et al., 2009; Cai et al., 2016b; Cai et al., 2019; Cai et al., 2014; LeBlond, 1978; Sassi and Hoitink, 2013). Thus, if the river discharge and mean sea

level remain constant, the variation of the mean slope with tidal forcing causes subtidal water levels to be higher at spring tide than at neap tide. The friction-induced modulation in subtidal water levels allows transporting the same volume of river water seaward over the neap-spring cycle (Guo et al., 2015). At settings with extended intertidal areas, the lateral spreading of the flood tidal wave produces additional frictional asymmetries between spring and neap tides that may also contribute to the increase the fortnightly tide amplitude (Friedrichs and Aubrey, 1988).

Without significant river discharge, fluctuations of the subtidal friction generated by tidal contributions alone may also produce a fortnightly tide (Vignoli et al., 2003). This case concerns many worldwide estuaries, typically in semi-arid regions, where the river flow influence is non-relevant compared to the tidal forcing during a large part of the year, at least (e.g., Correia et al., 2020; Descroix et al., 2020; Dias et al., 2016; Frota et al., 2013; Garel and D'Alimonte, 2017; Lamontagne et al., 2016; Lopez et al., 2020; McCutcheon et al., 2019; Valle-Levinson and Schettini, 2016). Due to high freshwater demands for

irrigation and other uses, semi-arid estuaries are under increasing stress worldwide from decreased freshwater inflows (Feng and Fu, 2013; Leblanc et al., 2012). As for tidal rivers, an accurate knowledge of water level variations at semi-arid estuaries with ephemeral freshwater inflows is necessary for ecosystem and freshwater management as well as for hydrodynamic research (Hoitink and Jay, 2016; Jay et al., 2011; Pan and Lv, 2019). However, the dynamics of the fortnightly tide produced by tidal asymmetry alone have been poorly documented so far. For example, the influence of tidal forcing at the mouth and

tidal wave reflection at the head or channel geometry on the MWL along the estuary is not sufficiently understood.

The present study proposes to make explicit the dynamics of fortnightly tides in estuaries where tidal motion is the main forcing. Subtidal water level observations in a semi-arid estuary with negligible freshwater discharge (the Guadiana) are compared with the outputs of analytical solutions considering a semi-diurnal tide of variable forcing amplitude propagating along a convergent channel. The objectives here are to establish the tidal properties that control the development of the subtidal



slope along the channel, to evaluate the effects of reflection at the head, and to explore MWL variations in function of the tidal

forcing and estuary geometry. In Sect. 2, the development of a fortnightly tide as a result of the balance between friction and

water slope is described analytically. The study site and collected data are presented in Sect. 3, along with the hydrodynamic

model used to reproduce tidal wave properties. The observational results are presented in Sect. 4. The model results are

presented and explored in Sect. 5 to elucidate and discuss the dynamics of the fortnightly tide and implications for estuarine

environments. The main conclusions of this work are summarized in Sect. 6.

## 2 Analytical description of the fortnightly tide

The subtidal water level slope along an estuarine channel produced by tidal effects can be derived from the 1D momentum
equation:

$$\frac{\partial U}{\partial t} + U\frac{\partial U}{\partial x} + g\frac{\partial Z}{\partial x} + g\frac{U|U|}{K^2 h^{4/3}} = 0, \qquad (1)$$

where $U$ is the cross-sectional average velocity, $h$ is the water depth, $Z$ is the water level fluctuation in relation to the tidally

average water level, $g$ is the gravitational acceleration, $t$ is time, $x$ is the longitudinal coordinate directed landward and $K$ is the

Manning-Strickler friction coefficient.

Assuming a periodic variation of flow velocity, the integration of Eq. 1 over a tidal cycle leads to (Cai et al., 2014; Vignoli et

al., 2003):

$$\frac{\partial \bar{Z}}{\partial x} = -\frac{1}{K^2}\overline{\left(\frac{U|U|}{h^{4/3}}\right)} - \frac{1}{2g}\frac{\overline{\partial U^2}}{\partial x}, \qquad (2)$$

where the overbars denote a tidal average. The second contribution to the subtidal water level in Eq. 2 originates from the

advective acceleration term, which is relatively small when compared to the first contribution induced by the residual frictional

effect as long as the Froude number is small (which is usually the case in estuaries, e.g., about 0.14 in the Guadiana; see also

Cai et al., 2019).

Thus, Eq. 2 simply expresses a balance between the slope and friction. In several cases, the contribution of the time-variable

depth is a second-order effect for the hydrodynamics (for instance, if the tidal amplitude is relatively small compared with the

depth), but it can still be relevant for the subtidal water level. The total free surface elevation $Z$ is the sum of its tidal average $\bar{Z}$

(responsible for the subtidal slope) and tidal wave height $Z_t$ such as $Z_t = Z - \bar{Z} = \zeta\bar{h}f_Z(t)$ where the function $f_Z$ describes the

variation of $Z$ with time and $\zeta$ is the ratio of the tidal amplitude to the water depth. From a Taylor expansion of $h = \bar{h} + Z_t$,

assuming that $\zeta \ll 1$, the frictional contribution to Eq. 2 can be rewritten as:

$$\frac{\partial \bar{Z}}{\partial x} = -\frac{1}{K^2\bar{h}^{4/3}}\overline{U|U|\left(1 - \frac{4}{3}\frac{Z_t}{\bar{h}}\right)}, \qquad (3)$$

where the additional term in parentheses can be significant, depending on the relative phase difference between $U$ and $Z$, i.e.,

the phase lead $\phi$ of the velocity with respect to the water level  (van Rijn, 2011). The analysis of a simple case like a purely

sinusoidal signal of frequency $\omega$ shows this feature clearly. If $U = v\cos(\omega t)$, where $v$ is the tidal velocity amplitude, and $f_Z =$




$\cos(\omega t - \phi)$, then $\overline{U|U|} = 0$, but $\overline{U|U|f_Z} = 4v^2 \cos(\phi)/(3\pi)$, which is vanishing only for $\phi = \pi/2 + n\pi$ (with $n$ being an integer number), i.e. for a standing wave. In the latter case, the peak of flow velocity occurs when the water level attains the mean sea level value both during the flood and the ebb phases (ebb discharge = flood discharge) and the tidally averaged water level keeps constant and horizontal along the channel. By contrast, if the wave contains a significant progressive component, the additional term produces a residual slope:

$$\frac{\partial \bar{Z}}{\partial x} = \frac{16}{9\pi} \frac{v^2 \zeta}{K^2 \bar{h}^{4/3}} \cos(\phi). \tag{4}$$

Equation 4 shows that a purely sinusoidal tidal wave may produce a residual slope even in the absence of freshwater discharge into the estuary. In this case, the residual slope relates to intratidal variations in the friction experienced by the wave for distinct tidal velocities (tidal stages). When the lag between elevation and velocity is significant, the peaks of ebb and flood velocities occur during low and high water stages, respectively. Thus, the frictional term yields larger dissipation during the ebb phase

than during the flood phase; this implies that the free surface slope must increase to compensate the dynamic imbalance in order to conserve a zero discharge. This mechanism is also stronger at spring than neap tides, resulting in fortnightly fluctuations of the residual slope; these dynamics are the scope of the present study.

## 3 Material and methods

### 3.1 Study area and data collection

#### 3.1.1 Study site overview

The Guadiana estuary consists of a single channel that connects the Guadiana River to the Gulf of Cadiz at the southern border between Portugal and Spain (Fig. 1). The channel is 78 km long and broadly oriented north-south, with a cross-section width that reduces exponentially from 700 m at the mouth to 60 m at the head. The thalweg is generally between 4 m and 8 m, with a mean depth of approximately 5.5 m (Garel, 2017).

Tides are semidiurnal in the region and their range falls in the microtidal to mesotidal regime (1.3 m during neaps and 2.6 m during springs, on average), with a maximum of 3.4 m. The amplitude of the dominant $M_2$ constituent varies by < 10% along the estuary. It is slightly damped along the lower half of the estuary, where friction dominates morphological convergence effects, and it is slightly amplified along the upper half, due to reflection at the head which overall effect is to reduce the friction experienced by the propagating wave (Garel and Cai, 2018).

The freshwater discharge into the estuary is generally low, in particular in summer when it is typically < 10 m³/s (see Garel and D'Alimonte, 2017). During a single tide, this rate corresponds to a volume of freshwater input approximately 70 times lower than the average tidal prism (30 Mm³; Correia et al., 2020). Under these conditions, the water column is generally well-mixed (Garel et al., 2009).



### 3.1.1 Data acquisition and processing

Water level variations along the Guadiana were measured from 31 July to 24 September 2015 with a series of 7 pressure transducers (Level TROLL 700 Data Logger, In-Situ), deployed every 10 km, approximately, from the mouth (St0) to 60 km upstream (St6; see Fig. 1). The sensor accuracy is rated at +/-0.55 cm by the manufacturer and their maximum range is 11 m. During the survey period, the mean freshwater discharge was 7 m$^3$/s and weather conditions were mild, typical of summers in the region. Equinoctial tides were on 24 August 2015 (neap) and 31 August 2015 (spring), with tidal ranges of 1.2 m and 3.3

m, respectively.

The raw pressure data, recorded at 1 min intervals, were smoothed with a 5 min moving average window and resampled to a common time at a 10 min interval. The records were then corrected for atmospheric pressure variations obtained from a station (Faro) located 50 km westward from the mouth (see Fig. 1 inset). The mean value was removed from the corrected time series to obtain water level variations around zero at each station. Pressure differences between Faro and Beja, located 110 km

northward (see Fig. 1 inset), were weak (< 2 mBar) during the survey, indicating an insignificant effect of meridional pressure variations on the water level slope along the estuary.

Tidal amplitudes at each station were obtained through demodulation of the pressure records (see Garel and Cai, 2018). The resulting time series were smoothed to discard jagged fluctuations induced by (small) diurnal inequalities of the astronomical tide. A Continuous Wavelet Transform (CWT) analysis was performed to compare the amplitude of tidal species (diurnal $D_1$,

semi-diurnal $D_2$, quarter-diurnal $D_4$ and fortnightly $D_f$) on spring and neap equinoctial tides at each station. The basic principles of CWT analyses are described in Jay and Flinchem (1997, 1999).

A Butterworth low-pass filter with a cut-off period of 11 days was applied to the time series to expose the fortnightly modulation of water level variations (denoted $Z_f$, hereafter). The low-pass filter was also applied with a 40-hr cut-off period (discarding shorter periodic variations, which are mainly tidal) in order to assess the contribution of $Z_f$ to the obtained subtidal

water level, $Z_s$. Significant differences between $Z_s$ and $Z_f$ are produced by external agents (such as atmospheric or coastal ocean processes) operating (and affecting the surface water level) in the subtidal to fortnightly band period.

In estuaries, a consistent increase of the MWL in the upstream direction is also produced by the horizontal water density gradient (e.g., Savenije, 2012; Savenije and Veling, 2005). Neap-spring variations in the salinity intrusion length may therefore contribute to the observed fortnightly water level modulation ($Z_f$). The effect of density on the slope was estimated based on

CTD measurements performed every ~4 km from the mouth to the freshwater front (defined as salinity <1 kg/g/m$^3$). These surveys were conducted at both high water slack (HWS) and low water slack (LWS) during consecutive spring (29 May 2018) and neap (06 June 2018) tides with range of 2.5 m and 1.1 m, respectively, under low river flow conditions (10 m$^3$/s). The tidally-averaged salinity curves were obtained as half the excursion length (Savenije, 2012). The contribution to the water level from the density effects (denoted $Z_\rho$) was estimated as (e.g., Cai et al., 2016a; Cai et al., 2019; Vignoli et al., 2003):

$$\frac{dZ_\rho}{dx} = -\frac{1}{2\rho_0} \overline{h_m \frac{d\rho}{dx}}, \tag{5}$$





where the axial distance $x=0$ at the mouth, $h_m$ is the measured cross-sectional mean depth and $\rho$ is the (depth-averaged) water density. Hereafter, the subscript zero indicates a value at the mouth.

## 3.2 Hydrodynamic model of tidal wave propagation

To derive the analytical solution for tidal hydrodynamics along an estuarine channel, it is assumed that the tidally-averaged cross-sectional area $\bar{A}$ and width $\bar{B}$ can be described by the following exponential functions (Savenije et al., 2008):

$$\bar{A} = \overline{A_0} \exp\left(-\frac{x}{a}\right),\tag{6}$$

$$\bar{B} = \overline{B_0} \exp\left(-\frac{x}{b}\right),\tag{7}$$

where $a$ and $b$ are the area and width convergence lengths, respectively. It is also assumed that the flow is concentrated in a rectangular cross-section, with a possible influence from storage areas described by the storage width ratio $r_S$ that is defined

as the ratio of the storage width $B_S$ to the tidally-averaged width $\bar{B}$ ($r_S = B_S/\bar{B}$). It directly follows from the assumption of a rectangular cross-section that the tidally-averaged depth is given by $\bar{h} = \bar{A}/\bar{B}$.

Toffolon and Savenije (2011) showed that the analytical solutions for tidal hydrodynamics along an estuary can be described by a few externally defined dimensionless parameters that depend on the estuary geometry and external forcing (see Table 1): the tidal amplitude $\zeta_0$, representing the boundary condition imposed at the estuary mouth; the estuary shape number $\gamma$,

indicating the effect of the channel cross-sectional area convergence; the friction number $\chi_0$, describing the role of the frictional dissipation at the estuary mouth; and the dimensionless estuary length $L_e^*$. In Table 1, $\eta_0$ is the tidal amplitude at the mouth, $c_0$ is the frictionless wave celerity in a prismatic channel (defined as $c_0 = \sqrt{g\bar{h}/r_S}$) and $L_0$ is the frictionless tidal wavelength ($L_0 = c_0/\omega$). The tidal frequency of the considered constituent is $\omega = 2\pi/T$ (with $T$ the tidal period). It is noted that the friction number $\chi_0$ is dependent of the Manning-Strickler friction coefficient $K$ that describes the effective friction resulting

from various environmental factors such as bedforms, grain roughness, vegetation, channel geometry (e.g., Savenije and Veling, 2005; Wang et al., 2014; Winterwerp and Wang, 2013), river discharge and from nonlinear effects induced by minor tidal constituents (Prandle, 1997). The value of $K$ is generally difficult to estimate accurately and is preferably obtained by calibrating the model results with observations, if available.

The corresponding dependent dimensionless parameters which are used to describe the main tidal hydrodynamics along the

channel include (Table 1): the actual tidal amplitude $\zeta$, the actual friction number $\chi$ ($\chi = 0$ in a frictionless case), the velocity number $\mu$ (the ratio of the actual velocity amplitude to the frictionless value in a prismatic channel), the celerity number for elevation $\lambda_A$ and velocity $\lambda_V$ (the ratio between the frictionless wave celerity in a prismatic channel and the actual wave celerity), the amplification number for elevation $\delta_A$ and velocity $\delta_V$ (describing the rate of increase, $\delta_A$ (or $\delta_V$) >0, or decrease $\delta_A$ (or $\delta_V$) <0 of the wave amplitudes along the estuary axis), and the phase lead angle $\phi$ between velocity $\phi_v$ and elevation $\phi_A$ ($\phi = 90°$

for a standing wave).

The dependent parameters defined in





Table 1 can be calculated using the equations developed by Toffolon and Savenije (2011) (see also, Cai et al., 2016b) for both infinite and semi-closed channels. To account for the longitudinal variation of the cross-sections in width and depth, the entire channel was subdivided into multiple reaches. The solutions were then obtained by solving a set of linear equations, with internal boundary conditions at the junction of the sub-reaches satisfying the continuity condition. Previous applications have shown that the model can accurately describe the effect of tidal forcing variations (e.g., spring and neap tides) on tidal properties along narrow convergent estuaries (such as the Guadiana) by considering a single effective tidal wave (Garel and Cai, 2018).

## 4 Observations

### 4.1 Fortnightly tide

Pressure records indicate large spring-neap fluctuations in tidal amplitude at the mouth (Fig. 2a), which are due to a relatively large $S_2$ constituent in the region (Garel and Cai, 2018). At the mouth (St0), there is no apparent relation between the tidal amplitude and the (11 day) low-passed filtered water level, $Z_f$. About 10 km upstream at St1, $Z_f$ occasionally accompanied the tidal forcing variations, in particular during the equinoctial tides on 24 (neap) and 31 (spring) August. Upstream (St2-6), $Z_f$ clearly co-varies with the tidal amplitude (Fig. 2a, b), being relatively higher at spring tide and lower at neap tide (Fig. 2b, c), typical of a fortnightly tide (e.g., Buschman et al., 2009; Hoitink and Jay, 2016; Sassi and Hoitink, 2013; Speer and Aubrey, 1985). The greatest water level variations (about 20 cm in range) are observed during equinoctial tides. It is also noted that $Z_f$ at St1 is incongruous in September, showing variations that are unrelated with the signal at adjacent stations.

Linear correlations confirm that the tidal forcing and fortnightly water level modulations are not correlated at the mouth (Fig. 3a, blue line). The correlation increases significantly upstream until St3 (where the coefficient of correlation $R$ is 0.8) and remains constant along the upper estuary half. Additionally, wavelet analyses show that during the equinoctial neap-spring tidal cycle the amplitude of the 15 days-period species ($D_f$) increased from about 0 cm at the mouth (St0) to 6 cm at St3 and upstream (Fig. 3b). Clearly, the fortnightly tide is produced in the lower estuary half and its amplitude upstream depends of the tidal forcing. It is also obvious in Fig. 2b that $Z_f$ remains constant along the upper estuary half, except during the first days of the survey ($Z_f$ increases along the entire channel); this is an artefact of the filtering process.

Despite their small amplitude (e.g., Fig. 3b), the fortnightly variations $Z_f$ contributed notably to the subtidal water level $Z_s$ at the upstream stations (compare Figs. 2b and 2c). For instance, the largest range of $Z_s$ variations at St3-6 occurred during the equinoctial tides. In the upper half of the estuary, $Z_s$ is largely controlled by the tidal forcing at the mouth, as indicated by the correlation between both parameters, which is marginally weaker than for $Z_f$ (Fig. 3a). During the survey, the residual $Z_s$-$Z_f$ (Fig. 2d), representing water level variations within a period band of 40 hours to 11 days, was associated with fluctuations in wind conditions at the mouth with periods of 7 and 9 days (not shown). The wind-induced water level variations are constant along the estuary and dominate the subtidal signal ($Z_s$) along its lower half. As the fortnightly tide grows in the upstream direction, $Z_f$ and wind-induced water level variations contributed similarly to $Z_s$ at the upper estuary half.





The above observations indicate that, under typical (fair) weather conditions that prevail in summer, the mean water level
along the Guadiana is affected by spring-neap variations produced within the system, characterized by an amplitude growth of
0.2 cm/km until ~30 km upstream. This distance is in the range of the salinity intrusion length for low river discharge conditions,
as exemplified by the salinity measurements performed in 2018 (Fig. 4). Despite expected differences in intrusion length at
high water slack (HWS) and low water slack (LWS), the tidally averaged (TA) salinity curves are highly similar on spring and
neap tides. From Eq. 5, the density-induced water level ($Z_\rho$) increases up to 5-6 cm from the mouth to the salt intrusion limit.
However, neap-spring differences in $Z_\rho$ over the fortnightly cycle are weak (~ 0.3 cm) along the channel. The density gradient
has therefore a negligible effect on the observed fortnightly modulation of water level variations.

To examine how the fortnightly signal is produced, the difference in $Z_f$ between St3 and St0 ($\Delta Z_f$, hereafter) is represented
along with the difference in high water level (HWL) and low water level (LWL) at the same stations ($\Delta$HWL and $\Delta$LWL,
respectively) for each tidal cycle (Fig. 5, with $\Delta Z_f$ on the right axis that is maximum on springs and minimum on neaps). We
note that these water level differences are not absolute values (since the pressure records were not referred to the same datum)
but rather indicate the temporal trend (i.e., increasing or decreasing) of the surface water slope at high and low water. For
instance, the difference in HWL between St3 and St0 varies weakly (Fig. 5, blue line), indicating that the slope at HWL is
relatively constant whatever the tidal forcing. In detail, $\Delta$HWL decreases (about 5 cm) from neap to spring tides, as opposed
to $\Delta Z_f$ variations (compare the blue and black lines in Fig. 5). By contrast, the relative difference in LWL between St3 and St0
(red line in Fig. 5) clearly features fortnightly variations (> 15 cm in range on equinoctial tides) which are in phase with $\Delta Z_f$.
This pattern indicates that the increase of the $Z_f$ slope (from neap to spring) between St0 and St3 is mainly produced by LWL
variations, as expected due to the strong depth dependence of frictional effects. The temporal variations of $\Delta$HWL and $\Delta$LWL
also indicate that the tidal wave height is significantly reduced at spring tide and slightly amplified at neap tide between St0
and St3. These differences in wave damping are examined in the next section considering equinoctial tides.

**4.2 Equinoctial tide propagation**

The observed equinoctial tidal waves on 24 (neap) and 31 (spring) August at Stations 0 to 6 are illustrated in Fig. 6. On both
dates, the wave is sinusoidal near the mouth (St0); as it propagates upstream, the wave remains approximately sinusoidal at
neap but is increasingly distorted at spring (the ebb and flood phases get typically longer and shorter, respectively).

The diurnal ($D_1$), semi-diurnal ($D_2$) and quarter-diurnal ($D_4$) species contained in the equinoctial tidal signal were extracted
with a wavelet analysis of the non-filtered pressure records (Fig. 7). The damping/shoaling patterns of these tidal species at
spring are similar to the ones of the main diurnal, semi-diurnal and quarter-diurnal tidal constituents along the estuary (see
Garel and Cai, 2018). The amplitude of $D_1$ is relatively low and constant along the channel on both neap and spring tides (Fig.
7a). $D_2$ is responsible for most of the tidal wave height variations along the channel (Fig. 7b), due to the predominance of the
$M_2$ constituent. On spring (neap), $D_2$ is damped (amplified) along the lower estuary half, while it is similarly amplified on both
spring and neap along the upper half. These along-channel variations in tidal elevation are noticeable in Fig. 6. The amplitude
of $D_4$ also features significant differences between spring and neap (Fig. 7c): it is weak (< 3 cm) and constant on neap, while





it grows upstream on spring from 4 cm at St0 up to approximately 20 cm at St6. The quarter-diurnal species consists mainly of $M_4$ (interaction of $M_2$ with itself) and $MS_4$ ($M_2$-$S_2$ interaction), both of equivalent importance in the Guadiana (Garel and Cai, 2018). Typically, the growth of these constituents in the upstream direction indicates increasing distortion of the tidal
wave due to non-linear frictional effects between HWL and LWL. Such along channel wave distortion is apparent on spring in Fig. 6b.

Amongst the three main tidal species at the Guadiana, $D_2$ is characterized by damping patterns that account for the previously reported fortnightly $Z_f$ modulations resulting from temporal LWL slope variations along the lower reach. There, the opposite semi-diurnal wave patterns on neap (amplification) and spring (damping) are due to variations in friction: frictional effects
dominate morphological convergence effects on spring tide and the opposite occurs on neap tide (Garel and Cai, 2018). By contrast, along the upper reach, where the fortnightly wave height is constant, $D_2$ amplifies similarly on springs and neaps due to reflection at the head that reduces the overall friction experienced by the (spring) wave (Garel and Cai, 2018). These observations indicate that the fortnightly tide amplitude along the channel is intrinsically linked, through friction, to the neap-spring variability of the damping patterns of the semi-diurnal wave propagating upstream. In the next section, the fortnightly
tide hydrodynamics are further explored with an analytical model considering a $M_2$ wave with variable amplitude.

## 5 Application of the analytical model

### 5.1 Model implementation

The hydrodynamic model of tidal propagation described in Sect. 3.2 was implemented at the Guadiana Estuary in order to compute the residual slope using Eq. 4. The model was set up with the same parameters as those in Garel and Cai (2018),
considering a flat bed and semi-closed channel with cross-section area convergence length (*a*) of 31 km and width convergence length (*b*) of 38 km (Eqs. 6 and 7), that accurately represents the estuary geometry. The model was calibrated with a Manning-Strickler coefficient $K$ of 47 $m^{1/3}$ $s^{-1}$. Considering a flat, non-rippled, sandy bed, $K$ is related to the drag coefficient ($C_d$) through (Soulsby, 1997):

$$C_d = \frac{g}{h^{1/3} K^2},$$ (8)

$$C_d = \left(\frac{0.4}{1 + \ln(Z_0/h)}\right)^2.$$ (9)

A typical value for the bed roughness in estuaries is $Z_0 = 0.7$ mm, corresponding to mixed (non-rippled) sand and mud surface sediment. Thus, for the mean water depth of 5.5 m considered here, the drag coefficient from Eq. 9 is $C_d = 2.5 \times 10^{-3}$. This coefficient is in the range of typical values applicable to depth averaged currents in estuaries, typically on the order of $2 \times 10^{-3} - 3 \times 10^{-3}$ (Dronkers, 2005; see also Li et al., 2004). This value yields a Manning-Strickler coefficient $K = 47$ $m^{1/3}$ $s^{-1}$, which
matches the value used to calibrate the model. The calibrated model reproduces remarkably well the observed properties





(amplitude and phase) of the spring and neap semi-diurnal tides along the channel when considering a M$_2$ wave of variable amplitude (see Garel and Cai, 2018).

To facilitate comparisons with observations, the mean value of the tidal average ($\bar{Z}$) time-series predicted by Eq. 4 was removed in order to obtain MWL variations around zero (denoted Z$_m$, hereafter), at each position along the channel (as for Z$_f$

at St0-6). Furthermore, the observed Z$_f$ at each station was corrected from the value at St0 to obtain Z$_f$ = 0 at the mouth (as for Z$_m$). Finally, the records at St1 were substituted by interpolated values between St0 and St2 to discard the incongruous observations reported in Sect. 4.1.

The model reproduces remarkably well the observed fortnightly tide (Fig. 8): Z$_m$ increases asymptotically along the lower estuary half only, with the largest amplitude (about 10-15 cm) on equinoctial tides. The tendency for the observed Z$_f$ to be

slightly higher (on spring) and lower (on neap) than Z$_m$ at St6 is attributed to backwater effects induced by a sill across the channel near the estuary head, not included in the model (see Garel and Cai, 2018). Overall, the correspondence of the analytical results with observations confirms that the fortnightly tide at the Guadiana results from intratidal variations in friction produced by distinct tidal stages.

## 5.2 Fortnightly tide dynamics

This section examines the relations between the fortnightly tide development and the main dimensional parameters that describe the M$_2$ tide propagation (Table 1): the damping number for water level, $\delta_A$ (i.e., the rate of increase, $\delta_A > 0$, or decrease, $\delta_A < 0$, of the wave amplitude along the estuary); the friction number $\chi$ (which represents the contribution of frictional dissipation); the celerity number for water level, $\lambda_A$ (i.e., the wave speed relatively to the frictionless wave celerity $c_0$, being < 1 for waves faster than $c_0$); and the phase difference $\phi$ between velocity and elevation. Both semi-closed and infinite channels

(denoted hereafter SC and IC, respectively) are considered to make explicit the effect of wave reflection at the head. Figure 9 represents the variations of the MWL slope ($S$), $\delta_A$, $\lambda_A$, $\chi$ and $\phi$ along the channel in function of the tidal forcing for both SC (a-e, left column) and IC (k-o, right column). To highlight the fortnightly variability of each parameter, the middle column (Fig. 9f-i) indicates their difference ($\Delta S$, $\Delta\delta_A$, $\Delta\lambda_A$, $\Delta\chi$ and $\Delta\phi$) considering the strongest (spring) and weakest (neap) tidal forcings (with SC as solid lines and IC as dashed lines).

The slope $S$ increases with tidal forcing along the lower half of the SC channel, but becomes insignificant upstream (Fig. 9a). Thus, the spring-neap difference in slope ($\Delta S$) becomes negligible at mid-estuary (Fig. 9f) and the fortnightly tide amplitude grows along the lower reach, as previously observed (Figs. 2a and 8a). By contrast, a significant slope $S$ develops along the entire infinite channel for tidal amplitudes larger than the mean value of 1 m, approximately (Fig. 9k). In this case, the fortnightly tide amplitude grows from the mouth to the head (see $\Delta S$ on Fig. 9f, dashed line). Such a pattern has been typically

observed in many long estuaries, although subject to substantial river discharge, where the MSf propagates much further landward that the semi-diurnal and diurnal tidal constituents (e.g. Buschman et al., 2009; Gallo and Vinzon, 2005; Godin, 1999; Godin and Martínez, 1994; Matte et al., 2014). In these settings, the river discharge effect can be accounted for by enhanced friction, increasing the tidal asymmetry in discharge, hence the fortnightly tide amplitude (Godin, 1985; Godin, 1999;





Laurel-Castillo and Valle-Levinson, 2020; Sassi and Hoitink, 2013; Savenije and Veling, 2005). Fortnightly oscillations of the

MWL eventually exceed the amplitude of the main tidal constituents, with mean LWL progressively being lowered during neap tides rather than spring tides (Gallo and Vinzon, 2005; LeBlond, 1979; LeBlond, 1991). Moreover, general observations of long systems indicate that the MSf amplitude maxima is located further landward than the maxima in the overtides (Gallo and Vinzon, 2005; Guo et al., 2015; Jay et al., 2015), which is not the case at the Guadiana (compare Figs. 3b and 7c). Overall, the analytical results show that wave reflection at the head inhibits the growth of the fortnightly tide along the upper reach of

estuaries.

Figures 9b and 9l illustrate how reflection affects the semi-diurnal wave amplitude variations along the channel. The wave shoals along the SC channel for tidal amplitude < 0.7 m (Fig. 9b). For larger tidal forcing, the wave is increasingly damped along the lower half of the channel but remains amplified upstream. Near the head, the wave shoals independently of the tidal forcing (as indicated by the verticalized isocontours in Fig. 9b). The spring-neap difference in damping $\Delta\delta_A$ is insignificant

along the upper estuary half (Fig. 9g, solid line), indicating that the difference in amplitude between the neap and spring wave remains constant (the wave is amplified in both cases). For an IC channel, the $M_2$ wave is damped along the entire system for tidal forcing > 0.6 m and amplified otherwise (Fig. 9l). Yet, damping is relatively weak, as indicated by the sub-horizontal contours on Fig. 9l. Thus, $\Delta\delta_A$ remains significant from the mouth to the head (Fig. 9g, dashed line).

Due to its nonlinear depth dependence, the friction number ($\chi$) reflects the above described variations of the $M_2$ wave height.

In particular, $\chi$ increases (decreases) with the tidal forcing or for shoaling (damped) waves (Fig. 9c, m). The frictional dissipation in the lower half of the estuary is similar for the SC and IF channels, indicating similar $M_2$ amplitude in both cases (Fig. 9b, l). By contrast, at the upper channel half, friction increases for SC and decreases for IF, in particular for large tidal forcing, due to opposite shoaling patterns. Hence, the greatest spring-neap differences in the friction term ($\Delta\chi$) between the SC and IF channels are in the upper 30 km (Fig. 9h). This pattern indicates a significantly higher spring wave for SC than IF along

this reach. Overall, the main effect of wave reflection at the head is to amplify the $M_2$ wave along the upper channel half, in particular for large tidal forcing. Compared to the IF case, these dynamics should enhance the MWL slope at spring, and thus the MSf amplitude at the upper reach, opposed to observations.

It can also be observed that for both the SC and IF channels, the $M_2$ wave accelerates with decreasing tidal forcing (Fig. 9d, n). Thus, the wave travels faster on neap tide than on spring tide. This pattern is explained by tidal damping effects on

progressive waves' celerity (Savenije et al., 2008; Savenije and Veling, 2005). As observed in many estuaries (e.g., the Thames, Shelde, Incomati), damped waves propagate slower than the classical wave celerity $c_0$ (corresponding to the frictionless case) while amplified waves often travel significantly faster than $c_0$. This phenomenon occurs because changes in the height of a propagating wave affect the phase difference $\phi$ between the horizontal and vertical tides (which is otherwise constant in the frictionless case). For IF channels, where damping is weak, the phase is relatively constant both spatially and temporally

(between 53° and 54.5°; Fig. 9o). The difference in phase between spring and neap ($\Delta\phi$) is < 2° (Fig. 9j, dashed line), indicating that the ebb-flood discharge asymmetry remains constant along the entire channel. As the wave height is only slightly damped, this asymmetry produces a significant MWL slope along the entire channel at spring. For the SC channel, the phase lead varies





significantly as the wave propagates upstream (see the vertical contours in Fig. 9e). This is because the tidal wave tends to
show a standing behaviour ($\phi \simeq 90°$) towards the head. The difference in phase between neap and spring $\Delta\phi$ is maximum at
the lower reach (up to 15°, representing a difference of about 50 min for a $M_2$ tide) but have a weak incidence on the fortnightly
slope which is slightly larger than for IF at this location (Fig. 9f). The main effect of reflection is in the upper reach where
HWL and LWL occur close to slack water, reducing the tidal discharge asymmetry. Hence, the discharge asymmetry is not
sufficient to produce a significant MWL slope on springs at the upper estuary half, although the wave is larger than for the IF
channel.

**5.2 Fortnightly tide amplitude and potential implications on estuarine environment**

In this section, the analytical model is exploited to analyse the fortnightly variations in MWL for SC channels with distinct
geometry (depth, length and morphological convergence). Implications for water resources management in semi-arid systems
are also discussed.

First, the maximum MWL ($\bar{Z}_{max}$) is predicted in function of the tidal forcing, using the analytical model that was setup for
the Guadiana (see Sect. 5.1). Overall, the tidal forcing largely controls $\bar{Z}_{max}$, which is significant (e.g. $> 0.5$ m) for macrotidal
($\eta > 2$ m), shallow, weakly convergent and long systems (Fig. 10a-c, respectively). Of the three geometric parameters
considered, the mean depth has the strongest influence on $\bar{Z}_{max}$ (Fig. 10a), as expressed in Eq. 4. The two other parameters
affect $\bar{Z}_{max}$ the most in the case of strongly convergent and short channels (as indicated by the sub-vertical contours in Fig.
10b, c), but the maximum MWL tends to remain small. For short systems, the superimposition of the incident and reflected
wave typically produces a standing wave (Cai et al., 2016b). In strongly convergent systems, the tide propagates as an 'apparent'
standing wave (Friedrichs and Aubrey, 1994; Hunt, 1964; Jay, 1991; Savenije et al., 2008; van Rijn, 2011). In both cases, the
tidal discharge asymmetry is negligible (the phase difference between velocity and elevation is close to 90°), inhibiting the
growth of $\bar{Z}_{max}$ along the channel.

Second, the position of $\bar{Z}_{max}$ along the channel is investigated for SC channels with usual geometric settings (i.e., estuary
length of 20-200 km, convergence length of 20-100 km and mean depth of 2-8 m). This position is defined at the distance from
the mouth where the slope becomes negligible ($< 0.1$ cm/km) and is denoted $X_{s=0}$. Results are reported in Fig. 11, considering
a tidal amplitude of 2 m at the mouth. Two geometric parameters are evaluated while the value of the 3[rd] parameter corresponds
to the one at the Guadiana (i.e., the convergence length is 38 km, the estuary length is 78 km and the water depth is 5.5 m in
Fig. 11a, b and c, respectively). In general, the slope gets negligible slightly upstream of the mid-estuary length. This position
varies weakly upstream or downstream with the convergence length and mean depth, except for deep and highly convergent
systems where $X_{s=0}$ is close to the mouth (Fig. 11b). In all the evaluated cases, the slope is flat along the upper estuary reach,
contrarily to infinite channels where $\bar{Z}_{max}$ is located at the estuary head.

The above results indicate that morphological changes in tidally-dominated estuaries with negligible river discharge affect
fortnightly water level variations along the channel. In particular, both an increased mean depth (due to channel dredging or





to sea level rise) and the installation of a tidal barrage (reducing the channel length) decrease the maximum fortnightly tide amplitude along the estuary (Fig. 10a, c). A greater depth reduces the friction of the propagating semi-diurnal tidal wave, and hence the MWL slope. Enhanced reflection effects by tidal barrages restrict the MWL growth to the first half of the (shorter) estuary, especially during spring tides. For both cases, the reduced growth of MSf amplitude implies weaker spring-neap differences in the LWL along the channel (e.g., Fig. 5). Margins usually exposed to air on spring low tide may experience

permanent inundation in the modified estuary. Such changes in the tidal inundation regime can have severe impacts on estuarine environments. In particular, tidal inundation is a fundamental driver of wetland functions, and even small changes can influence the extent and function of saltmarsh habitats (Janousek and Folger, 2014; Valiela et al., 1978). An increased LWL also enhances flooded areas during strong river discharge events, which are predicted to increase both in intensity and frequency in semi-arid regions (e.g., Smith, 1996; Tabari, 2020). Moreover, in comparison with (diurnal and semi-diurnal)

tidal components that dominate at the mouth, the upstream mass transport by MSf motions is more effective due to a longer wavelength (Buijsman and Ridderinkhof, 2007; MacMahan et al., 2014). In particular, variations in subtidal water level are directly linked to salt intrusion into estuaries (Henrie and Valle-Levinson, 2014). Reductions of the fortnightly tide amplitude in modified estuaries may therefore have significant effects on water quality and ecology. These potential effects on the fortnightly tide are particularly relevant for macrotidal semi-arid estuaries like those found in NE Brazil (e.g., Barletta and

Costa, 2009; Clark and Pessanha, 2014; Dias et al., 2009; Frazão and Vital, 2006).

**6 Conclusions**

This study has examined the fortnightly water level variations due to tidal motions alone in tide-dominated estuaries. These systems are typically found in semi-arid regions, where the river discharge is negligible during a large part of the year. In the Guadiana estuary, pressure measurements in August-September 2015 show that the mean water level along the estuary is

typically higher on spring tide and lower on neap tide. These fluctuations result from an increase of the relative LWL in the upstream direction on spring tides, while the relative HWL remains approximatively constant. The MSf amplitude grows from the mouth until the mid-estuary and remains constant upstream. During the survey, weather conditions were fair, and the fortnightly tide represented about half of the subtidal signal at the upper estuary half. The other main contribution, with approximately constant amplitude from the mouth to the head, was induced by wind (not studied further in the present paper).

The contribution of neap-spring differences in water density is negligible.

It is confirmed that the fortnightly tide is produced by intratidal variations in friction using an analytical model of wave propagation. Frictional asymmetries relate mainly to the nonlinear depth dependence of friction to water depth which strongly (weakly) affects the balance between ebb and flood discharges on springs (neaps). Considering a semi-closed channel, the model results match the observations, indicating that the MSf growth in the first half of the estuary is due to reflection effects

at the head. Reflection affects MWL mainly through modification of the phase difference between velocity and elevation, which increases in the upstream direction (the wave is standing near the head due to the superimposition of the incident and

reflected waves). The low and high water levels get progressively closer to slack water, reducing the flood-ebb discharge asymmetry. In the upper half of the estuary, the discharge asymmetry becomes negligible, despite wave shoaling. Overall, observations of a flat MWL along a significant portion of the upper estuary, in particularly on springs, may indicate the

presence of significant reflection effects. These data are generally easier to obtain than the phase difference, which requires combined velocity and elevation records.

Finally, changes in the mean depth (e.g. due to dredging or sea level rise) or in the channel length (e.g., due to the installation of a tidal barrage) affect the MSf amplitudes along semi-arid estuaries with negligible river discharge. Impacts on the ecosystem may arise from the induced modification of the inundation regime (through change of the LWL) and of the upstream

mass transport, in particular in macrotidal regions.

**Author contributions**

All authors contributed to the design and development of the work. The experiments were originally carried out by HC. PZ carried out the data analysis. EG built the model and wrote the paper.

**Competing interests**

The authors declare that they have no conflict of interest.

**Acknowledgments**

EG acknowledges the support of the Portuguese Foundation for Science and Technology (FCT) through the grant UID/MAR/00350/2020 attributed to CIMA, University of Algarve. HC acknowledges the support of the National Natural Science Foundation of China (grant no. 51979296). Authors would like to thank Prof. Marco Toffolon from Trento University,

who raised the issue of fortnightly water level dynamics in tide-dominated estuaries.

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

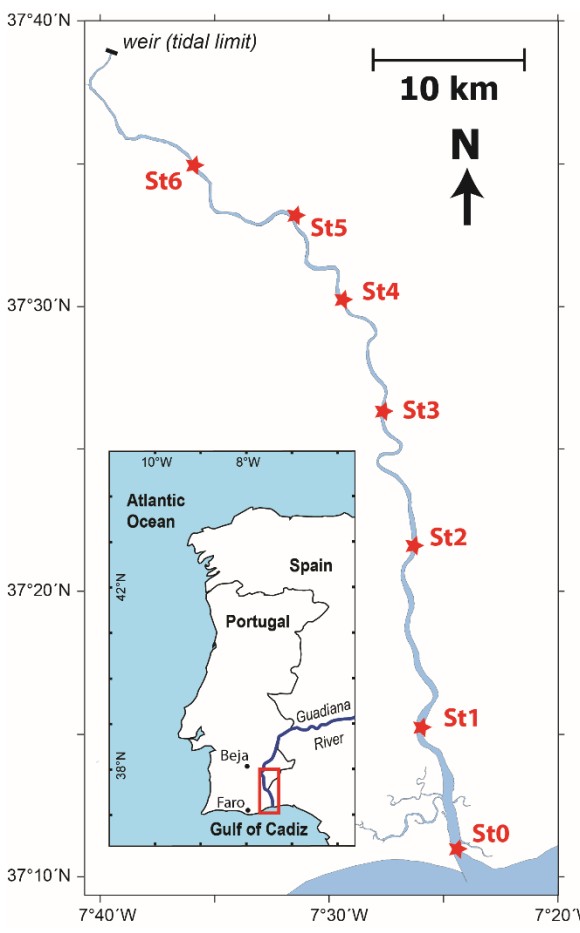

**Figure 1: Location of the pressure measurement stations along the Guadiana Estuary (St0-6, red stars) and general location (inset).**







Figure 2: (a) Tidal amplitude ($\eta_0$, m) near the mouth (St0). (b) low-pass filtered (11 days) water level ($Z_f$, m) at stations St0-6 along the estuary. (c) low-pass filtered (40 hr) water level ($Z_s$, m) at St0-6. (d) residual water level $Z_s$- $Z_f$.




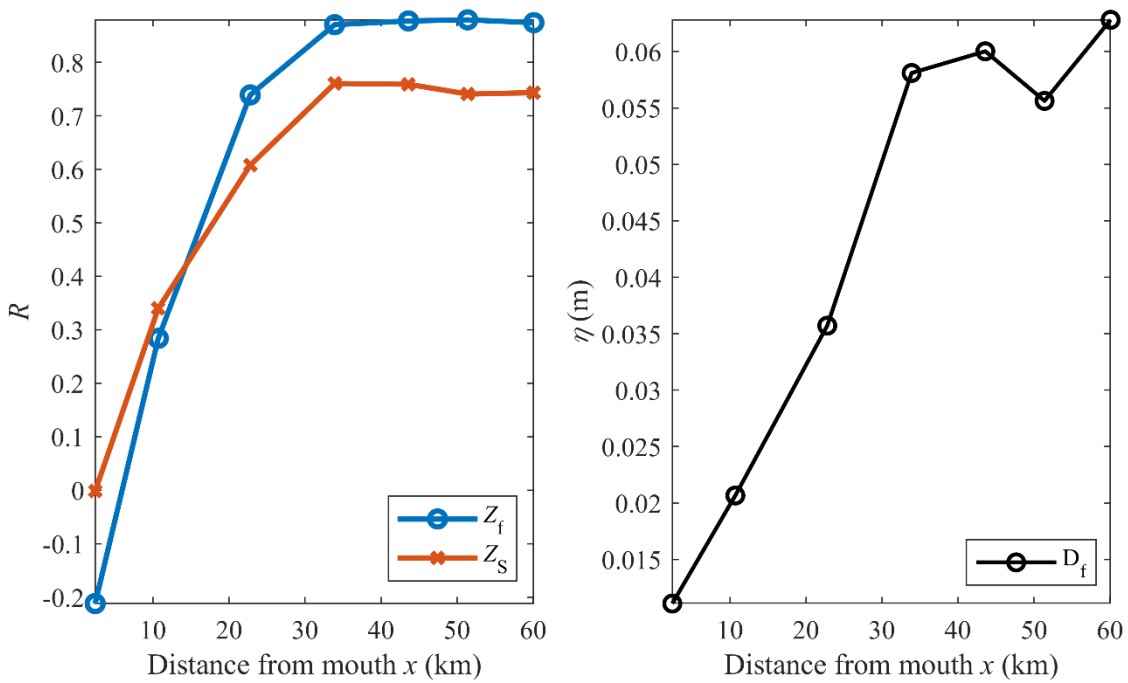

Figure 3: (a) Correlation coefficient ($R$) of the tidal amplitude forcing with the fortnightly water level $Z_f$ (blue line) and subtidal water level $Z_S$ (red line) along the estuary. (b) Amplitude of the 15 days tidal species $D_f$ along the estuary during the equinoctial neap-spring tidal cycle from 24-31 August 2015.

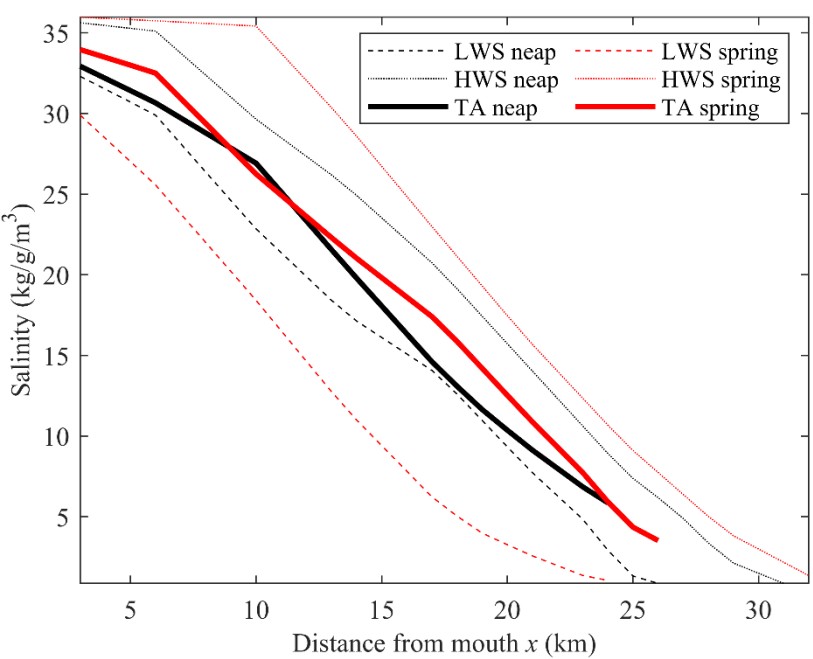

**Figure 4: Vertically averaged salinity along the Guadiana estuary at spring tide (29 May 2019, red lines) and neap tide (06 June 2018, black lines) under a river discharge of 10 m³/s. The sampling was performed at high water slack (HWS, dotted lines) and low water slack (LWS, dashed lines) every 2-4 km along the channel. TA (solid lines) represents the tidally averaged salinity.**


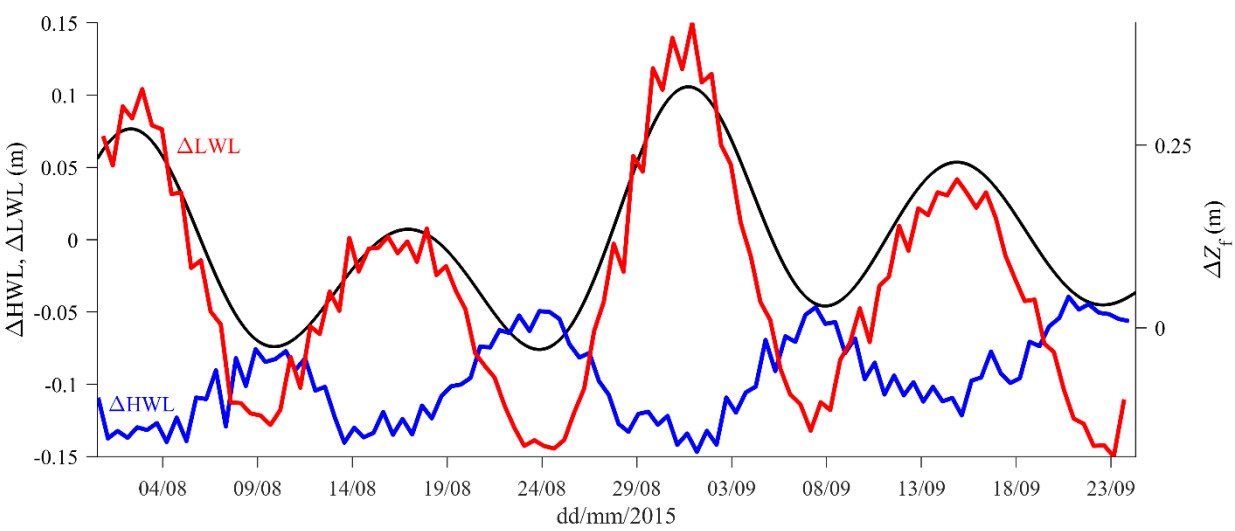

**Figure 5: Differences in the relative water level between St3 and St0: fortnightly tide ($\Delta Z_f$, black line, right axis), $\Delta$HWL (blue line, left axis) and $\Delta$LWL (red line, left axis).**




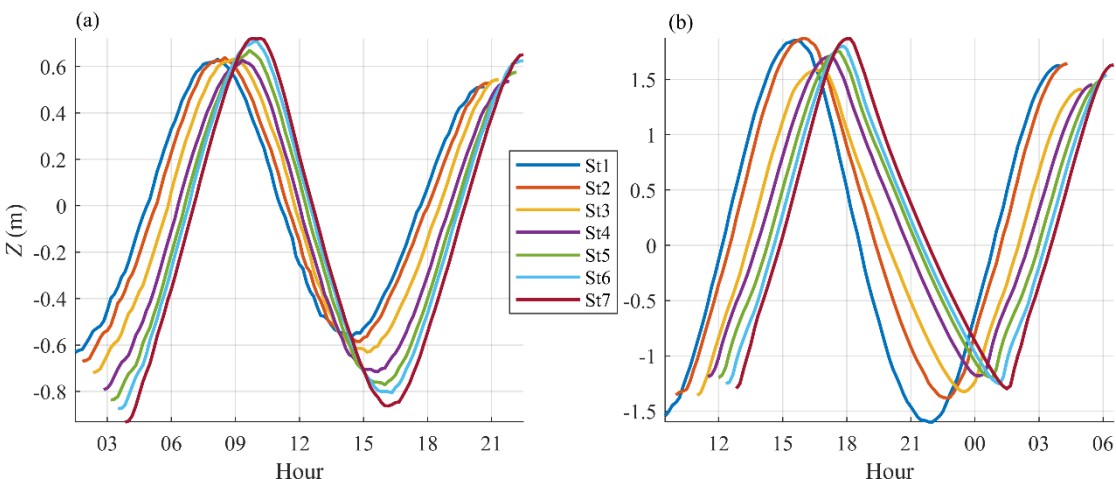

**Figure 6: Water level variations (Z, m) at stations St0-6 at (a) neap (24/08/2015) and (b) spring (31/08/2015) equinoctial tides.**

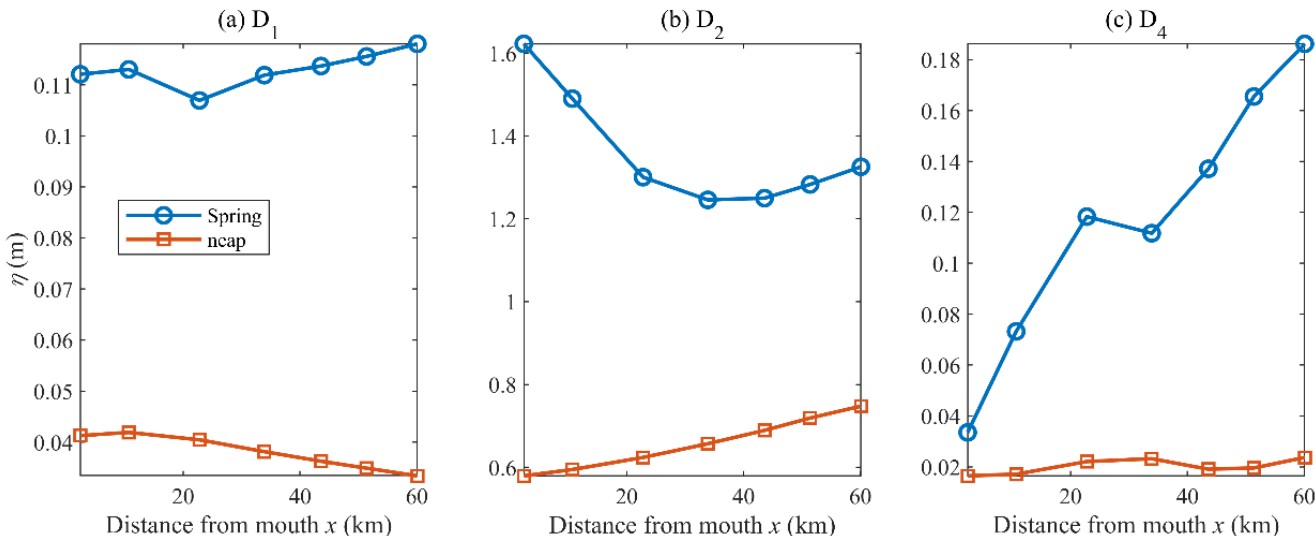

**Figure 7: Amplitude ($\eta$, m) of tidal species (a) diurnal $D_1$, (b) semi-diurnal $D_2$ and (c)quarter-diurnal $D_4$ along the channel during equinoctial spring (blue line) and neap (red line) tides.**




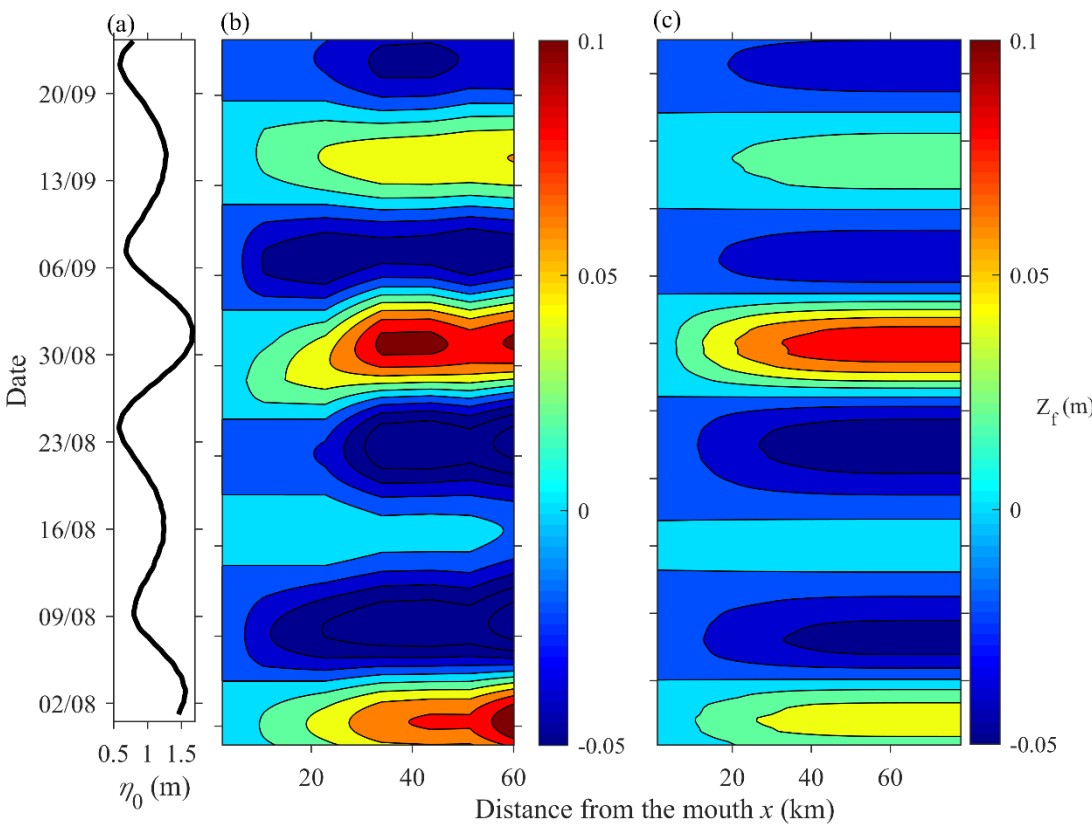

**Figure 8: (a) Tidal amplitude at the mouth ($\eta_0$, m), (b) observed ($Z_f$) and (c) simulated ($Z_m$) fortnightly water level variations along the Guadiana estuary.**









**Figure 9: Tidally averaged slope ($S$) and main dimensionless parameters describing the tidal propagation along the Guadiana in function of the tidal amplitude at the mouth ($\eta_0$, m): the damping number for water level ($\delta_A$), the friction number ($\chi$), the celerity number for water level ($\lambda_A$) and the phase angle between velocity and elevation ($\phi$). The left column (a-e) and the right column (k-o) represent the results for a semi-closed channel and an infinite channel, respectively; the middle column (f-j) represents the spring-neap difference for each parameter ($\Delta S$, $\Delta \delta_A$, $\Delta \lambda_A$ and $\Delta \phi$, respectively). The thick red lines presented in subplots b and l correspond to $\delta_A=0$; in subplots d and n, they correspond to $\lambda_A=1$.**



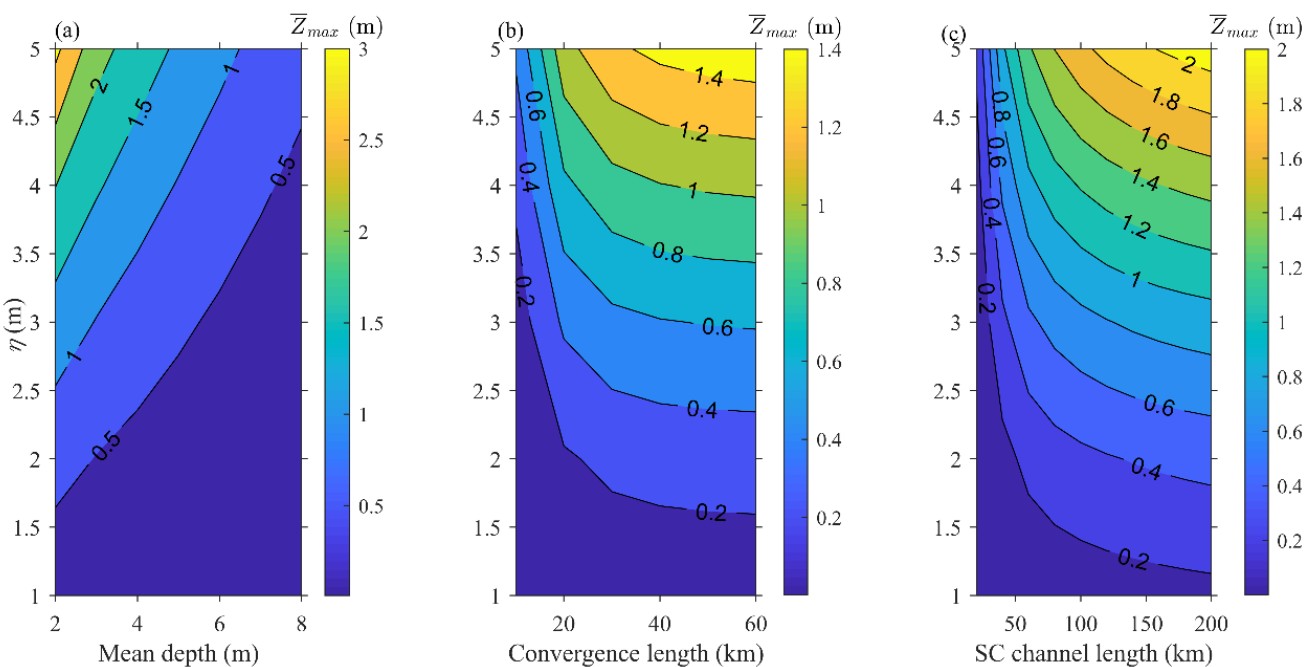


**Figure 10: Maximum tidally average water level ($\overline{Z}_{max}$) as a function of the tidal forcing and mean depth (a), convergence length (b) and estuary length (c) at a semi-closed channel.**




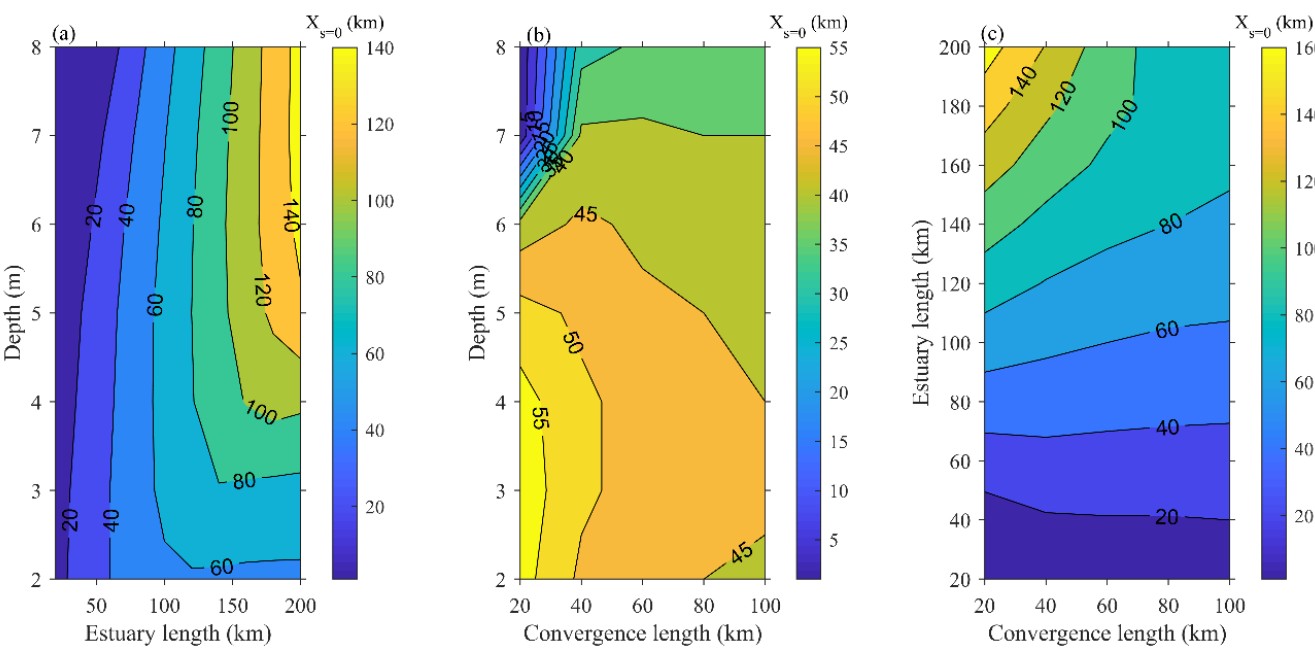


**Figure 11: Position of negligible slope ($X_S$=0, in km) along a semi-closed channel with distinct geometric settings. In (a) the convergence length is 38 km, in (b) the estuary length is 78 km and in (c) the water depth is 5.5 m (as at the Guadiana).**







**Table 1: The definitions of dimensionless parameters.**

| Independent | Dependent |
|---|---|
| | Tidal amplitude $\zeta = \eta / \bar{h}$ |
| | Friction number $\chi = r_S c_0 \zeta g / \left( K^2 \omega \bar{h}^{4/3} \right)$ |
| Tidal amplitude at the mouth $\zeta_0 = \eta_0 / \bar{h}$ | Velocity number $\mu = \upsilon / \left( r_S \zeta c_0 \right) = \upsilon \bar{h} / \left( r_S \eta c_0 \right)$ |
| Estuary shape number $\gamma = c_0 / \left( \omega a \right)$ | Celerity number for water level $\lambda_A = c_0 / c_A$ |
| Friction number at the mouth $\chi_0 = r_S c_0 \zeta_0 g / \left( K^2 \omega \bar{h}^{4/3} \right)$ | Celerity number for velocity $\lambda_V = c_0 / c_V$ |
| Estuary length $L_e^* = L_e / L_0$ | Damping /amplification number for water level $\delta_A = c_0 \, \mathrm{d}\eta / \left( \eta \omega \, \mathrm{d}x \right)$ |
| | Damping number for velocity $\delta_V = c_0 \, \mathrm{d}\upsilon / \left( \upsilon \omega \, \mathrm{d}x \right)$ |
| | Phase lead angle $\phi = \phi_V - \phi_A$ |