# Peer review of "Dynamics of fortnightly water level variations along a tide-dominated estuary with negligible river discharge"

_Ocean Science, 2021_

## Author Comment (AC1)

**Response letter to Reviewer#1**

We thank Reviewer#1 for the careful consideration of our work. We agree with his/her constructive and thoughtful comments and suggestions, which led to a much improved and complete manuscript. In this response letter, we have replied (in blue) to all the comments formulated by the Reviewer (in black).

**Comments:**

This manuscript discusses fortnightly (~14.8 day) variations in surface elevation along an estuary, as recorded in measurements and found in a model. The focus is on tidal forcing with neglect of river discharge and atmospheric forcing. The introduction and discussion indicate that these elevation variations can be significant in several contexts. However, given considerable existing literature, some with river discharge, this is a simpler scenario and the results are in general unsurprising. Good agreement between model and observations is obtained. The degree of novelty can be questioned and one would expect more discussion of the relation with previous results including river discharge (which I guess has varied magnitude relative to the tidal prism in the various contexts treated in the literature).

Our reply: We very much appreciate all the comments raised by the reviewer. In the revised manuscript, we shall completely address all the comments. Specifically, we shall highlight the novelty of this manuscript in the introduction:

"Overall, this research proposes, for the first time, an analytical tool for assessing the impacts of geometric changes (such as the mean depth or length of the estuary) on fortnightly water level variations in tide-dominated estuaries with negligible river discharge. The results shed new light on how fortnightly dynamics in water levels is generated due to imposed tidal forcing at the mouth and tidal wave reflection from the head of the estuary. Through estimates of the MWL along the estuary, the approach is specifically helpful for sustainable water resources management of ecosystems, especially in macrotidal estuaries."

1. Non-dimensional parameters are introduced but not sufficiently exploited in my opinion. There is a focus on values pertaining to the observed Guadiana estuary. What about an estuary near resonance with M2?

Our reply: Indeed, the proposed analytical model is mainly used to better understand the fortnightly water level dynamics along the Guadiana, an example of semi-arid estuaries which have been poorly documented so far. Given the fact that we have reproduced the tidal hydrodynamics (including resonance behavior) in the Guadiana estuary by means of an analytical model in our previous work (Garel and Cai, 2018), here in this study, we did not discuss more details with regard to resonance behavior. 2. I am not convinced by the (repetitive rather than justified) attribution of the behavior to friction rather than non-linearity in the advection terms. This must depend on their relative magnitude which depend on the length and depth of the estuary. I question the analysis in 4.2. Non-linear effects on  $M_2$  usually show in  $M_4$  as a result of advection and  $M_6$  as a result of quadratic friction. However, here the quarter-diurnal species (mainly  $M_4$ ) are attributed mainly to friction and sixth-diurnal species are not discussed.

Our reply: We agree with the reviewer's comments on the generations of the even harmonics (e.g., M4) and odd harmonics (e.g., M6). Indeed, the generation of quarterdiurnal species is mainly attributed to the nonlinear continuity term, the convective acceleration term and the quadratic friction term, while the generation of sixth-diurnal species is primarily induced by the quadratic friction term (Parker, 1991). In the revised manuscript, we shall modify the corresponding sentence as: *"Typically, the growth of these constituents in the upstream direction indicates increasing distortion of the tidal wave due to the combined effects from non-linear continuity term, convective acceleration term and quadratic friction in both mass and momentum equations (Parker, 1991)"*.

In addition, we shall explicitly mention that:

"Note that the sixth-diurnal species were not analysed since their magnitudes are relatively small when compared with other components (see Garel and Cai, 2018)."

3. In general the English is good and understandable, despite occasional strange usage which should be picked up in copy-editing. However

There is frequent reference to "equinoctial" which is incorrect and obscures the intended meaning, e.g. lines 199, 239, 240, 241, 244, 596. Omit and say precisely the time referred to.

Our reply: We thanks the reviewer to point this out. In the revised manuscript, we shall discard "equinoctial tides". The manuscript now refers to the largest and lowest tides occurring during the study period, which will be explicitly mentioned in section 3.1.1.: "*The largest spring and lowest neap tides were on 31 August 2015 and 24 August 2015, with tidal ranges of 3.3 m and 1.2 m, respectively.*"

 "spring" and "neap" should be used as adjectives e.g. "spring tide(s)" or perhaps as plurals e.g. "springs". Do not use "spring" alone; this is a season of the year. Typos etc.

Our reply: We agree with this comment. In the revised manuscript, we shall modify the expressions concerning "spring" and "neap".

4.

■ Lines 9, 49. The direction of increase should be stated.

Our reply: You are right! In the revised manuscript, we shall modify the sentences as: "Observations indicate that the fortnightly fluctuations in mean amplitude of water level increase in the upstream direction along the lower half of a tide-dominated estuary (the Guadiana) with negligible river discharge but remain constant upstream."

"At settings with extended intertidal areas, the lateral spreading of the flood tidal wave produces additional frictional asymmetries between spring and neap tides that may also contribute to the increase of the fortnightly tide amplitude in the upstream direction (Friedrichs and Aubrey, 1988)."

■ Line 34. "metric order" should be "of order 1 m"? Our reply: Corrected as suggested.

■ Equation (1) and line 77. There should be a reference for this form of friction, especially the depth-dependence.

Our reply: In the revised manuscript, we shall explicitly include the reference "Savenije, 2012", especially for the depth-dependence in the friction term.

• Line 83. The relative magnitude of the two terms on RHS(2) is independent of |U| so the relevance of Froude number is unclear.

Our reply: Actually, more details with regard to the relevance of Froude number can be found in Cai et al. (2019):

"Note that the contribution from advective acceleration to the residual water level slope

$$\frac{\partial \overline{Z_{adv}}}{\partial x} = -\frac{1}{2g} \frac{\partial \overline{U^2}}{\partial x}$$
(1.1)

can be easily integrated to

$$\overline{Z_{adv}} = -\frac{1}{2g} \left( \overline{U^2} - \overline{U_0^2} \right) = -\frac{1}{2} \overline{Fr_0} \left( \frac{\overline{U^2}}{\overline{U_0^2}} - 1 \right) \overline{h_0}$$
(1.2)

where the subscript 0 indicates the values at the estuary mouth. Here the Froude number is introduced,  $\overline{Fr^2} = \overline{U^2} / (g\overline{h})$ , which is computed with the averaged variables. In this case, the correction is local (not cumulative) and proportional to the flow depth through a coefficient that is negligible as long as the velocity does not change significantly, and *Fr* is small, as is common in most tidal flows."

■ Line 129. "Equinoctial" can be omitted: not strictly true and made redundant by the other information given.

Our reply: We agree with this comment. In the revised manuscript, we shall remove "Equinoctial tides".

■ Line 138. Why should a diurnal tide induce "jagged" fluctuations? Specify their

time scale and that of the smoothing.

Our reply: The jagged fluctuations in tidal range are induced by diurnal inequalities of the (semi-diurnal) tide which produce small differences in the height of two successive tides (see figure A below). In the revised manuscript, we shall explicitly mention that: *"The resulting amplitudes were smoothed using a 6-point moving average to discard jagged fluctuations induced by (small) diurnal inequalities of the astronomical tide."*

*Figure A. Tidal amplitude (m) at the estuary mouth (St0): non filtered data are in black, smoothed data are in red.*

Lines 139-141. If another scientist were to check this work, they would need more specification of the CWT. Also line 187 "using the equations developed by." is too vague. Either they should be repeated or at least equation numbers in the cited papers should be specified.

Our reply: We agree with the reviewer's comments. In the revised manuscript, we shall provide more details with regard to these two methods:

*"The basic principles of CWT analyses are described in Jay and Flinchem (1997; see Eqs. 1-3 in their manuscript)."*

"The dependent parameters defined in Table 1 can be calculated using the equations developed by Toffolon and Savenije (2011; see also Cai et al., 2016 and Eqs. 6-20 in Garel and Cai, 2018) for both infinite and semi-closed channels."

■ Line 190. "Continuity" of volume flux? Surface elevation? Anything else? Our reply: In the revised manuscript, we shall explicitly mention that:

"The solutions were then obtained by solving a set of linear equations, with internal boundary conditions at the junction of the sub-reaches satisfying the continuity conditions for both water level and discharge."

 Line 202. This sentence is unclear. Which period shows the greatest variations? "equinoctial" occurs at the very end of the record. Likewise, lines 212-213.
Our reply: In the revised manuscript, we shall clarify that:

"The greatest water level variations of  $Z_f$  (about 20 cm in range) correspond to the largest changes in tidal height, observed between the 24 and 31 August 2015 (see Fig. 2b)."

"For instance, the largest range of Zs variations at St3-6 occurred during the spring and neap tides on the 24-31 August 2015."

■ Lines 245-246. This sentence seems tautological; what exactly is being compared with what?

Our reply: In the revised manuscript, we shall remove this sentence since we used the same data set.

■ Lines 273-275. This is not correctly expressed. (8) relates K to Cd but (9) is an independent value for Cd, presumably coming from Souls by (1997).

Our reply: Here the main purpose of adopting Eqs. (8) and (9) is to justify the calibrated Manning-Strickler friction coefficient.

■ Line 337. "opposed to observations" begs discussion which I don't see. Our reply: We shall clarify the meaning of this sentence:

"Compared to the IF case, these dynamics should enhance the MWL slope at springs, and thus the MSf amplitude at the upper reach, opposed to observations (see Fig. 2)."

■ Line 350. 15° is about 31 minutes for M2. Our reply: Corrected as suggested.

■ Line 355. This is section 5.3! Our reply: Corrected as suggested.

■ Figures 7, 8. It is better practice to include zero on the amplitude scale. Our reply: Many thanks for the suggestions. In the revised manuscript, we shall modify the Figure 7a-c and Figure 8a.

Figure 7: Amplitude  $(\eta, m)$  of tidal species (a) diurnal  $D_1$ , (b) semi-diurnal  $D_2$  and (c)quarter-diurnal  $D_4$  along the channel during the largest spring (blue line, 31 August 2015) and lowest neap (red line, 24 August 2015) tides.

---

## Author Response (AR1)

**Response letter to Reviewer#1**

We thank Reviewer#1 for the careful consideration of our work. We agree with his/her constructive and thoughtful comments and suggestions, which led to a much improved and complete manuscript. In the revised paper, we have replied (in blue) to all the comments formulated by the Reviewer (in black). The line numbers in this rebuttal refer to the revised version of the manuscript.

**Comments:**
This manuscript discusses fortnightly (~14.8 day) variations in surface elevation along an estuary, as recorded in measurements and found in a model. The focus is on tidal forcing with neglect of river discharge and atmospheric forcing. The introduction and discussion indicate that these elevation variations can be significant in several contexts. However, given considerable existing literature, some with river discharge, this is a simpler scenario and the results are in general unsurprising. Good agreement between model and observations is obtained. The degree of novelty can be questioned and one would expect more discussion of the relation with previous results including river discharge (which I guess has varied magnitude relative to the tidal prism in the various contexts treated in the literature).

Our reply: We very much appreciate all the comments raised by the reviewer. In the revised manuscript, we have completely addressed all the comments. Specifically, we have highlighted the novelty of this manuscript in the introduction:
*"Overall, this research proposes, for the first time, an analytical tool for assessing the impacts of geometric changes (such as the mean depth or length of the estuary) on fortnightly water level variations in tide-dominated estuaries with negligible river discharge. The results shed new light on how fortnightly dynamics in water levels is generated due to imposed tidal forcing at the mouth and tidal wave reflection from the head of the estuary. Through estimates of the MWL along the estuary, the approach is specifically helpful for sustainable water resources management of ecosystems, especially in macrotidal estuaries."*
*(Page 3, lines 66-69)*

1. Non-dimensional parameters are introduced but not sufficiently exploited in my opinion. There is a focus on values pertaining to the observed Guadiana estuary. What about an estuary near resonance with $M_2$?

Our reply: Indeed, the proposed analytical model is mainly used to better understand the fortnightly water level dynamics along the Guadiana, an example of semi-arid estuaries which have been poorly documented so far. Given the fact that we have reproduced the tidal hydrodynamics (including resonance behavior) in the Guadiana estuary by means of an analytical model in our previous work (Garel and Cai, 2018),

here in this study, we did not discuss more details with regard to resonance behavior.

2. I am not convinced by the (repetitive rather than justified) attribution of the behavior to friction rather than non-linearity in the advection terms. This must depend on their relative magnitude which depend on the length and depth of the estuary. I question the analysis in 4.2. Non-linear effects on $M_2$ usually show in $M_4$ as a result of advection and $M_6$ as a result of quadratic friction. However, here the quarter-diurnal species (mainly $M_4$) are attributed mainly to friction and sixth-diurnal species are not discussed.

Our reply: We agree with the reviewer's comments on the generations of the even harmonics (e.g., $M_4$) and odd harmonics (e.g., $M_6$). Indeed, the generation of quarter-diurnal species is mainly attributed to the nonlinear continuity term, the convective acceleration term and the quadratic friction term, while the generation of sixth-diurnal species is primarily induced by the quadratic friction term (Parker, 1991). In the revised manuscript, we have modified the corresponding sentence as:
*"Typically, the growth of these constituents in the upstream direction indicates increasing distortion of the tidal wave due to the combined effects from non-linear continuity term, convective acceleration term and quadratic friction in both mass and momentum equations (Parker, 1991)".*
*(Page 9, lines263-265)*

In addition, we have explicitly mentioned that:
*"Note that the sixth-diurnal species were not analysed since their magnitudes are relatively small when compared with other components (see Garel and Cai, 2018)."*
*(Page 9, lines254-255)*

3. In general the English is good and understandable, despite occasional strange usage which should be picked up in copy-editing. However

■ There is frequent reference to "equinoctial" which is incorrect and obscures the intended meaning, e.g. lines 199, 239, 240, 241, 244, 596. Omit and say precisely the time referred to.
Our reply: We thank the reviewer to point this out. In the revised manuscript, we have discarded "equinoctial tides". The manuscript now refers to the largest and lowest tides occurring during the study period, which are explicitly mentioned in section 3.1.1.:
*"The largest spring and lowest neap tides were on 31 August 2015 and 24 August 2015, with tidal ranges of 3.3 m and 1.2 m, respectively."*
*(Page 5, lines134-135)*

■ "spring" and "neap" should be used as adjectives e.g. "spring tide(s)" or perhaps as plurals e.g. "springs". Do not use "spring" alone; this is a season of the year. Typos etc.
Our reply: We agree with this comment. In the revised manuscript, we have modified the expressions concerning "spring" and "neap".

4.

■ Lines 9, 49. The direction of increase should be stated.

Our reply: You are right! In the revised manuscript, we have modified the sentences as:
*"Observations indicate that the fortnightly fluctuations in mean amplitude of water level increase in the upstream direction along the lower half of a tide-dominated estuary (the Guadiana) with negligible river discharge but remain constant upstream."*
*(Page 1, lines9-11)*

*"At settings with extended intertidal areas, the lateral spreading of the flood tidal wave produces additional frictional asymmetries between spring and neap tides that may also contribute to the increase of the fortnightly tide amplitude in the upstream direction (Friedrichs and Aubrey, 1988)."*
*(Page 2, lines48-50)*

■ Line 34. "metric order" should be "of order 1 m"?

Our reply: Corrected as suggested.

■ Equation (1) and line 77. There should be a reference for this form of friction, especially the depth-dependence.

Our reply: In the revised manuscript, we have explicitly included the reference "Savenije, 2012", especially for the depth-dependence in the friction term.

■ Line 83. The relative magnitude of the two terms on RHS(2) is independent of |U| so the relevance of Froude number is unclear.

Our reply: Actually, more details with regard to the relevance of Froude number can be found in Cai et al. (2019):
"Note that the contribution from advective acceleration to the residual water level slope

$$\frac{\partial \overline{Z_{adv}}}{\partial x} = -\frac{1}{2g}\frac{\partial \overline{U^2}}{\partial x} \tag{1.1}$$

can be easily integrated to

$$\overline{Z_{adv}} = -\frac{1}{2g}\left(\overline{U^2} - \overline{U_0^2}\right) = -\frac{1}{2}\overline{Fr_0}\left(\frac{\overline{U^2}}{\overline{U_0^2}} - 1\right)\overline{h_0} \tag{1.2}$$

where the subscript 0 indicates the values at the estuary mouth. Here the Froude number is introduced, $\overline{Fr^2} = \overline{U^2}/\left(g\overline{h}\right)$, which is computed with the averaged variables. In this case, the correction is local (not cumulative) and proportional to the flow depth through a coefficient that is negligible as long as the velocity does not change significantly, and *Fr* is small, as is common in most tidal flows."

■ Line 129. "Equinoctial" can be omitted: not strictly true and made redundant by

the other information given.

Our reply: We agree with this comment. In the revised manuscript, we have removed "Equinoctial tides".

■ Line 138. Why should a diurnal tide induce "jagged" fluctuations? Specify their time scale and that of the smoothing.

Our reply: The jagged fluctuations in tidal range are induced by diurnal inequalities of the (semi-diurnal) tide which produce small differences in the height of two successive tides (see figure A below). In the revised manuscript, we have explicitly mentioned that: *"The resulting amplitudes were smoothed using a 6-point moving average to discard jagged fluctuations induced by (small) diurnal inequalities of the astronomical tide." (Page 5, lines143-145)*

[Figure]

*Figure A. Tidal amplitude (m) at the estuary mouth (St0): non filtered data are in black, smoothed data are in red.*

■ Lines 139-141. If another scientist were to check this work, they would need more specification of the CWT. Also line 187 "using the equations developed by." is too vague. Either they should be repeated or at least equation numbers in the cited papers should be specified.

Our reply: We agree with the reviewer's comments. In the revised manuscript, we have provided more details with regard to these two methods:

*"The basic principles of CWT analyses are described in Jay and Flinchem (1997; see*

*Eqs. 1-3 in their manuscript)."*
*(Page 5, lines147-148)*
*"The dependent parameters defined in Table 1 can be calculated using the equations developed by Toffolon and Savenije (2011; see also Cai et al., 2016b and Eqs. 6-20 in Garel and Cai, 2018) for both infinite and semi-closed channels."*
*(Page 7, lines194-195)*

■ Line 190. "Continuity" of volume flux? Surface elevation? Anything else?

Our reply: We have explicitly mentioned that:

*"The solutions were then obtained by solving a set of linear equations, with internal boundary conditions at the junction of the sub-reaches satisfying the continuity conditions for both water level and discharge."*
*(Page 7, lines196-198)*

■ Line 202. This sentence is unclear. Which period shows the greatest variations? "equinoctial" occurs at the very end of the record. Likewise, lines 212-213.

Our reply: In the revised manuscript, we have clarified that:

*"The greatest water level variations of $Z_f$ (about 20 cm in range) correspond to the largest changes in tidal height, observed between the 24 and 31 August 2015 (see Fig. 2b)."*
*(Page 7, lines208-210)*
*"For instance, the largest range of $Z_s$ variations at St3-6 occurred during the neap-spring cycle on 24-31 August 2015."*
*(Page 8, lines219-220)*

■ Lines 245-246. This sentence seems tautological; what exactly is being compared with what?

Our reply: In the revised manuscript, we have removed this sentence since we used the same data set.

■ Lines 273-275. This is not correctly expressed. (8) relates K to Cd but (9) is an independent value for Cd, presumably coming from Souls by (1997).

Our reply: Here the main purpose of adopting Eqs. (8) and (9) is to justify the calibrated Manning-Strickler friction coefficient.

■ Line 337. "opposed to observations" begs discussion which I don't see.

Our reply: We had clarified the meaning of this sentence:

*"Compared to the IF case, these dynamics should enhance the MWL slope at springs, and thus the MSf amplitude at the upper reach, opposed to observations (see Fig. 2)."*
*(Page 12, lines345-346)*

■ Line 350. 15° is about 31 minutes for M2.

Our reply: Corrected as suggested.

■ Line 355. This is section 5.3!

Our reply: Corrected as suggested.

■ Figures 7, 8. It is better practice to include zero on the amplitude scale.

Our reply: Many thanks for the suggestions. In the revised manuscript, we have modified the Figure 7a-c and Figure 8a.

[Figure]

*Figure 7: Amplitude ($\eta$, m) of tidal species (a) diurnal $D_1$, (b) semi-diurnal $D_2$ and (c)quarter-diurnal $D_4$ along the channel during the largest spring (blue line, 31 August 2015) and lowest neap (red line, 24 August 2015) tides.*

[Figure]

**Figure 8: (a) Tidal amplitude at the mouth ($\eta_0$, m), (b) observed ($Z_f$) and (c) simulated ($Z_m$) fortnightly water level variations along the Guadiana estuary.**

■ Figure 8. The scales for distance from the mouth differ between (b) and (c) and do not include zero.

Our reply: In the revised manuscript, we have modified the Figure 8 (see the figure above).

■ Table 1 expressions for damping numbers: if dη/dx and dv/dx are factors then they need to appear without other symbols interspersed.

Our reply: We had modified the expressions of Damping /amplification number for water level and Damping/amplification number for velocity in Table 1:

*"Damping /amplification number for water level"*:

$$\delta_A = \frac{c_0}{\eta \omega} \frac{\mathrm{d}\eta}{\mathrm{d}x}$$

*"Damping/amplification number for velocity"*:

$$\delta_V = \frac{c_0}{\upsilon \omega} \frac{\mathrm{d}\upsilon}{\mathrm{d}x}$$

**Response letter to Reviewer#2**

We thank the Reviewer#2 for the positive evaluation of our work and for the constructive and thoughtful comments and suggestions which led to an improved manuscript. In the revised paper, we have addressed all the comments formulated by the Reviewer by replying (in blue) to his/her remarks (in black). The line numbers in this rebuttal refer to the revised version of the manuscript.

**Comments:**

The manuscript by Garel et al. is a very focused piece of research on the fortnightly variability in mean water level along a tide-dominated estuary with low river inflow. They apply an analytical model to determine the physical processes involved, validating their results against observations in the Guadiana estuary. The presentation of the research is well structured, has a good coverage of supporting literature to define existing knowledge - identifying the gaps to address, and has good use of figures to illustrate key points.

Our reply: Thanks a lot for the positive assessment of our manuscript. We very much appreciate all the comments raised by the Reviewer. In the revised manuscript, we have completely addressed all the comments.

1. I have a few minor comments, with the main one focusing on the implications of the research. Section 5.2 answers the initial research question posed, but it is not until this point (near to the end of the manuscript) that the reader has a good understanding of the importance of this research in terms of application to present-day management issues. More detail is required earlier in the manuscript, for example L19 in the abstract more information about the impacts could be stated. On L56/57 more information is given

but this could be summarized in the abstract. "Impacts on the estuarine environment" is very general and could be a positive or negative impact and may relate to the natural system and/or human influence on the system. While ecosystems are mentioned in the conclusions, flood hazard is not.

Our reply: We very much appreciate the Reviewer's comments. In this study, the potential impacts on the estuarine environment is mainly due to the changes in terms of upstream mass transport and inundation regime rather than the flood hazard, since we mainly focus on a tide-dominated estuary with negligible river discharge. In the abstract part, we have explicitly mentioned that:

*"This has significant potential impacts on the estuarine environment in terms of ecosystem management."*
*(Page 1, lines19-20)*

In the conclusions part, we also have explicitly mentioned that:
"*Impacts on the ecosystem may arise from the induced modification of the inundation regime (through changes of the MWL and LWL) and of the upstream mass transport, in particular in macrotidal regions.*"
*(Page 14, lines428-430)*

2. In the paragraph starting at L131, the influence of atmospheric pressure has been considered and removed. Could you comment on the influence that wind and waves may have on the water levels. I appreciate winds are mentioned later in the manuscript but a comment to acknowledge the limitations of not removing all meteorological (and wave) forcing on the water level would be of value.

Our reply: Indeed, the other meteorological forcing (e.g., wind and wave) may also exert on a considerable impact during a storm surge event. In the revised manuscript, we have explicated mentioned that:
*"It should be noted that the potential impacts induced by wind and waves on the water level dynamics during storms are not addressed since the study focuses on normal (fair) meteorological conditions.*"
*(Page 5, lines141-142)*

3. I have a few other minor suggestions:
■  L34, "metric order", I suggest providing a value the change in elevations may reach.
Our reply: In the revised manuscript, we have replaced "metric order" with "of order 1 m".

■  L35, "flood control" I would have thought flood hazard was related to the changes in mean high water rather than mean water levels. This point isn't discussed later in 5.2. Are you really referring to inundation of the intertidal and the impacts on ecosystems as discussed later in section 5.2 or potential changes in saltmarsh that protect the coast from flooding? Please ensure the points in the introduction are directly linked to the discussion points in section 5.2 and vice versa.

Our reply: You are right! Here, we would like to emphasis the "tidal inundation" of the intertidal area at high tide rather than the "flood hazard" related to the high water levels and high river discharges. In the revised manuscript, we have replaced "flood control" with "tidal inundation".

■ L206 & L244, "species" is an unusual choice of term. I would have used component.

Our reply: We would prefer to keep "tidal species". The CWT is not able to distinguish tidal constituents (such as $M_2$), but groups of constituents with similar frequencies (such as semi-diurnal, $D_2$). A group of tidal constituents with similar frequencies is commonly denoted as a tidal species (see for example Buschman et al., 2009, Hoitink and Jay, 2016, Jay, 1991, Jay and Flinchem, 1997, Sassi et al., 2012).

■ L209, Zf remains at a similar level but it is not perfectly constant.

Our reply: We agree with this comment. In the revised manuscript, we have modified the sentence as:

*"It is also obvious in Fig. 2b that $Z_f$ remains at a similar level along the upper estuary half."*
*(Page 8, lines217)*

■ L210, it would be better to focus on the period where there are quality results stating the initial data are removed due to artefacts of the filtering techniques.

Our reply: We have followed this recommendation. It is now mentioned in the manuscript:

"*The initial 4 days of the time-series were discarded due to artefacts produced by the filtering process.*"
*(Page 5, lines152)*
The axes of Figures 2, 5 and 8 were updated accordingly.

■ L235, both lines have a clear signal. One is larger than the other rather than clearer.
Our reply: In the revised manuscript, we have modified the sentence as:
*"By contrast, the relative difference in LWL between St3 and St0 (ΔLWL, red line in Fig. 5) was larger than ΔHWL (> 15 cm in range between the 24 and 31 August), and clearly features fortnightly variations in phase with Δ$Z_f$."*
*(Page 8, lines242-244)*

■ L272, how realistic is the use of a flat non-ripped sandy bed. How could the results vary for different conditions?

Our reply: We have not considered a flat non-rippled sandy bed to obtain the *K* parameter. Typically, *K* was obtained by comparing the model results with observations. In this paragraph, we intend to check if our *K* value is realistic or not. Calculations based on equations 8 and 9 indicate that the calibrated *K* corresponds to a drag coefficient (Cd) which is in the range of typical values applicable to depth averaged currents in estuaries.

- L396, clarify the study is numerical using a case study for validation. I suggest you start the conclusion saying, "in a numerical representation of tide-dominated estuaries". Otherwise, it could be misinterpreted that the results are only applicable to the study site.

Our reply: Many thanks for the suggestion. In the revised manuscript, we have modified the sentence as:

*"Using analytical solutions, this study has examined the fortnightly water level variations due to tidal motions alone in tide-dominated estuaries with negligible river discharge."*

*(Page 14, lines406-407)*

4. There are a few typos to correct including:

- L16 and L46 need rewording for clarity.

Our reply: We sincerely thank for pointing these out. In the revised manuscript, we have modified the sentences as:

*"Observations of a flat mean water level along a significant portion of an upper estuary suggest a standing wave character and thus indicate significant reflection of the propagating semi-diurnal wave at the head."*

*(Page 1, lines15-17)*

*"The friction-induced modulation in subtidal water levels allows transporting, for any tidal amplitude, the same volume of riverine water seaward over the neap-spring cycle (Guo et al., 2015)."*

*(Page 2, lines47-48)*

- L34, check punctuation.

Our reply: We have modified the sentence as:

*"Subtidal changes in water elevation at these systems can be of order 1 m at the upper reaches and may have as such significant effects on navigability and tidal inundation (e.g., Aubrey and Speer, 1985; Godin, 1999; Guo et al., 2015; Jay et al., 2015; Matte et al., 2014)."*

*(Page 2, lines34-37)*

- L46, check gramma.

Our reply: We have modified the sentence as:

*"The friction-induced modulation in subtidal water levels allows transporting, for any tidal amplitude, the same volume of riverine water seaward over the neap-spring cycle (Guo et al., 2015)."*

*(Page 2, lines47-48)*

- P6/P7, there is an unexpected break in the sentence across the pages.

Our reply: Corrected as suggested.

■ L205, insert a space between "is 0.8"

Our reply: Corrected as suggested.

■ L208, depends on.

Our reply: Corrected as suggested.

■ L250, nearly constant.

Our reply: Corrected as suggested.

■ L403, in the upper estuary.

Our reply: In the revised manuscript, we have replaced "at the upper estuary" with "in the upper estuary".

■ In the Conclusion both abbreviations (e.g. LWL) and full names (e.g. mean water level) are used. Be consistent.

Our reply: Thanks a lot for pointing this out. In the revised manuscript, we have modified the full name (e.g. mean water level) into abbreviations for the whole manuscript.

References:

Aubrey, D.G., and Speer, P.E.: A study of non-linear tidal propagation in shallow inlet/estuarine systems Part I: Observations, Estuar. Coastal Shelf S., 21, 185-205, doi: 10.1016/0272-7714(85)90097-6, 1985.

Buschman, F.A., Hoitink, A.J.F., van der Vegt, M., and Hoekstra, P.: Subtidal water level variation controlled by river flow and tides, Water Resour. Res., 45, W10420, doi: 10.1029/2009WR008167, 2009.

Cai, H., Toffolon, M., and Savenije, H. H. G.: An Analytical Approach to Determining Resonance in SemiClosed Convergent Tidal Channels, Coast Eng. J., 58, doi: Artn 1650009 10.1142/S0578563416500091, 2016b.

Cai, H., Savenije, H.H.G., Garel, E., Zhang, X., Guo, L., Zhang, M., Liu, F., and Yang, Q.: Seasonal behaviour of tidal damping and residual water level slope in the Yangtze River estuary: identifying the critical position and river discharge for maximum tidal damping, Hydrol. Earth Syst. Sci., 23, 2779-2794, doi: 10.5194/hess-23-2779-2019, 2019.

Friedrichs, C.T., and Aubrey, D.G.: Non-linear tidal distortion in shallow well-mixed estuaries: a synthesis, Estuar. Coast. Shelf S., 27, 521-545, doi: 10.1016/0272-7714(88)90082-0, 1988.

Garel, E., and Cai, H.: Effects of Tidal-Forcing Variations on Tidal Properties Along a Narrow Convergent Estuary, Estuar. Coast., 41, 1924-1942, doi: 10.1007/s12237-018-0410-y, 2018.

Godin, G.: The Propagation of Tides up Rivers with Special Considerations on the Upper Saint Lawrence River, Estuar. Coast. Shelf S., 48, 307-324, doi: 10.1006/ecss.1998.0422, 1999.

Guo, L., van der Wegen, M., Jay, D.A., Matte, P., Wang, Z.B., Roelvink, D., and

He, Q.: River-tide dynamics: Exploration of nonstationary and nonlinear tidal behavior in the Yangtze River estuary, J. Geophys. Res-Oceans, 120, 3499-3521, doi: 10.1002/2014JC010491, 2015.

Hoitink, A.J.F., and Jay, D.A.: Tidal river dynamics: Implications for deltas, Rev. Geophys., 54, 240-272, doi: 10.1002/2015RG000507, 2016.

Jay, D.A.: Green's law revisited: Tidal long-wave propagation in channels with strong topography, J. Geophys. Res-Oceans, 96, 20585-20598, doi: 10.1029/91JC01633, 1991.

Jay, D.A., and Flinchem, E.P.: Interaction of fluctuating river flow with a barotropic tide: A demonstration of wavelet tidal analysis methods, J. Geophys. Res-Oceans, 102, 5705-5720, doi: 10.1029/96JC00496, 1997.

Matte, P., Secretan, Y., and Morin, J.: Temporal and spatial variability of tidal-fluvial dynamics in the St. Lawrence fluvial estuary: An application of nonstationary tidal harmonic analysis, J. Geophys. Res-Oceans, 119, 5724-5744, doi: 10.1002/2014JC009791, 2014.

Sassi, M.G., and Hoitink, A.J.F.: River flow controls on tides and tide-mean water level profiles in a tidal freshwater river, J. Geophys. Res-Oceans, 118, 4139-4151, doi: 10.1002/jgrc.20297, 2013.

Parker, B. B.: The relative importance of the various nonlinear mechanisms in a wide range of tidal interactions, in: Tidal Hydrodynamics, edited by: Parker, B., John Wiley and Sons, Hoboken, N. J., 237–268, 1991.